# [Re] Learning to count everything

## Reproducibility Summary

**Scope of Reproducibility**

The core finding of the paper is a novel architecture FamNet for handling the few-shot counting task. We examine its implementation in the provided code on GitHub and compare it to the theory in the original paper. The authors also introduce a data set with 147 visual categories FSC-147, which we analyze. We try to reproduce the authors' results on it and on CARPK data set. Additionally, we test FamNet on a category specific data set JHU-CROWD++. Furthermore, we try to reproduce the ground truth density maps, the code for which is not provided by the authors.

**Methodology**

We use the combination of the authors' and our own code, for parts where the code is not provided (e.g., generating ground truth density maps, CARPK data set preprocessing). We also modify some parts of the authors' code so that we can evaluate the model on various data sets. For running the code we used the Quadro RTX 5000 GPU and had a total computation time of approximately 50 GPU hours.

**Results**

We could not reproduce the density maps, but we produced similar density maps by modifying some of the parameters. We exactly reproduced the results on the paper's data set. We did not get the same results on the CARPK data set and in experiments where implementation details were not provided. However, the differences are within standard error and our results support the claim that the model outperforms the baselines.

**What was easy**

Running the pretrained models and the demo app was quite easy, as the authors provided instructions. It was also easy to reproduce the results on a given data set with a pretrained model.

**What was difficult**

It was difficult to verify the ground truth density map generation as the code was not provided and the process was incorrectly described. Obtaining a performant GPU was also quite a challenge and it took quite many emails to finally get one. This also meant that we were unable to reproduce the training of the model.

**Communication with original authors**

We contacted the authors three times through issues on GitHub. They were helpful and responsive, but we have not resolved all of the issues.

# 1    Introduction

Counting objects in a scene is a task that is very simple and intuitive for humans, however, the problem arises when there are hundreds, thousands, or even more objects in one scene as the counting becomes difficult or impossible. Yet, sometimes it is beneficial to have a count estimation of such big amounts of objects and that is why many approaches for counting objects have been proposed. These methods can easily outperform humans, especially when there are many objects in a scene. Still, the advantage of humans is that we are able to count objects from the majority of visual categories with ease, which is not the case with the current object counting methods. In fact, the counting approaches that have been proposed until now can usually handle only one visual category at the time, and even those categories are mostly limited to a few, most frequently humans [1, 4, 5, 13, 18], vehicles [3, 6, 11, 12, 19], and animals [2, 20]. The reason behind these limitations in the currently proposed approaches is twofold. The majority of counting approaches requires dot annotations for thousands of objects on few thousands of training images. The second reason is that there exists no large enough unconstrained data set, which would allow the development of a method for counting any visual category. Both of these limitations exist as dot annotation and development of a large enough data is a laborious and a costly task.

In this report we try to reproduce the paper *Learning to Count Everything* [14], in which the authors try to overcome both of the above mentioned limitations. Instead of mimicking the previous works and treating counting as a fully supervised regression task, they pose counting as a few shot regression task. This approach is generalizable as only an input image with a few exemplars from the same image (that represent the object of interest) is required to achieve generalization to a completely novel visual category class. Second, the authors of this paper also address the lack of data sets with many visual categories as they introduce a data set including more than 6000 images from 147 visual categories.

# 2    Scope of reproducibility

The authors are interested in counting everything and they achieve that by posing counting as a few-shot regression task. The core finding of the paper is a novel architecture called FamNet that handles a few-shot counting task together with a novel adaptation strategy that adapts the network to any novel visual category at test time, by using only a few exemplar objects from the novel category. Furthermore, the authors introduce a data set containing 147 different visual categories and they show that their method outperforms other state-of-the-art approaches – object detectors as well as few-shot counting approaches. We test these key findings from the paper:

- FamNet outperforms other few-shot approaches when it comes to object counting.

- FamNet performs well even on a category-specific data set.

- Increasing the number of exemplars decreases FamNet's error.

# 3    Methodology

Where available, we use the authors' code from GitHub. We modify it so that we can evaluate the model on different data sets. Additionally, we prepare our scripts for generating ground truth density maps, ablation study, and preprocessing of CARPK data set, as the authors do not provide it.

### 3.1    Model descriptions

FamNet is composed of two main modules – a multi-scale feature extraction module and a density prediction module. The multi-scale feature extraction module is based on the ImageNet pretrained network, more specifically on the first four blocks from a pretrained ResNet-50 backbone. From the code, we find out that the authors use the pretrained ResNet-50 model from TorchVision. The density prediction module is designed in a way to be agnostic to the visual categories. They achieve this by not feeding the features obtained from the feature extraction module directly. Contrary, they rather use the correlation map between the exemplar features and image features as the input to the density prediction module. We show a visualization of inputs to the density prediction module in Appendix A.

As mentioned, the proposed FamNet can adapt to a new visual category once trained, using only a few exemplars. To understand the novel adaptation loss that is used during test time we first quickly describe the Min-Count and Perturbation losses.

Let B denote the set of provided exemplar bounding boxes (bounding boxes denoting examples of the object, that we are counting, given to the network). For each bounding box $b \in B$, let $Z_b$ represent the crop from the density map $Z$ at location $b$.

### 3.1.1 Min-Count Loss

Min-Count Loss is defined as

$$L_{\text{MinCount}} = \sum_{b \in B} \max(0, 1 - ||Z_b||_1). \tag{1}$$

The idea behind this loss is that the sum of density values within $Z_b$ should be at least 1 as the predicted count is a sum of predicted density values, and there is at least one object at the location $b$. Meaning that if the total value of the density map in the exemplar box is equal to or greater than 1, the loss will not increase for this location, but if the total value of the density map in the exemplar box is smaller than 1, we increase the loss.

By inspecting the authors' code, however, we find out that Min-Count loss is incorrectly implemented. Instead of using the difference between 1 and $||Z_b||_1$, the authors use the squared difference. In notation, the implementation of Min-Count loss in the original implementation is

$$L_{\text{MinCount}}^{\text{implemented}} = \sum_{b \in B} \max(0, (1 - ||Z_b||_1)^2). \tag{2}$$

We address the issue and test the performance of the model for both implementations in Section 4.

### 3.1.2 Perturbation Loss

Perturbation Loss is defined as

$$L_{\text{Per}} = \sum_{b \in B} ||Z_b - G_{h \times w}||_2^2, \tag{3}$$

where $G_{h \times w}$ is a 2D Gaussian window of size $h \times w$ and standard deviation $\sigma_G = 8$. The authors do not provide the reasoning for the chosen value, so we try different options to investigate its influence. We report our findings in Section 4.2.1. This loss is inspired by the success of tracking algorithms based on correlation filter, where algorithms learn a filter that has the highest response at the location of the bounding box and lower response at all perturbed locations. We can look at the density map $Z$ as the correlation response between the exemplars and the image.

### 3.1.3 Adaptation loss

The final loss, called adaptation loss, is defined as a weighted combination

$$L_{\text{Adapt}} = \lambda_1 L_{\text{MinCount}} + \lambda_2 L_{\text{Per}}, \tag{4}$$

where $L_{\text{MinCount}}$ is the Min-Count Loss, $L_{\text{Per}}$ is Perturbation Loss and $\lambda_1$ and $\lambda_2$ are scalar hyper-parameters. The authors fine-tuned them on validation set, and we use the same values $\lambda_1 = 10^{-9}$ and $\lambda_2 = 10^{-4}$. Note that adaptation loss is only used at test time, and MSE between predicted and ground truth density map over all pixels is used as a loss during training.

## 3.2 Data sets

### 3.2.1 FSC-147

As the majority of the data sets are dedicated to a specific visual category, the authors collected and annotated 6135 images across 147 different visual categories. The average image height is 774 and the average image width is 938 pixels. In each image, all objects are dot-annotated in an approximate center of the object. Furthermore, in a majority of cases (96.26%) three object instances are randomly selected and are additionally annotated with axis-aligned bounding

boxes denoting exemplar bounding boxes. In some cases four (3.45%), five (0.27%), or six (0.02%) object instances are additionally annotated.

The data set is divided into train, validation, and test sets in a way that each of these sets does not share any object categories. The train, validation, and test sets consist of 3659, 1286, and 1190 images, respectively.

The authors provide two sets of ground truth density maps which are the same in all but two cases (3417.npy and 3477.npy). In the second set of ground truth density maps, the first image appears more blurred, while different objects are counted on the latter image (see Figure 1).

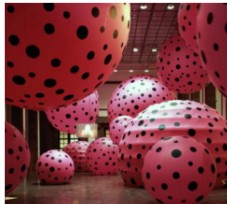 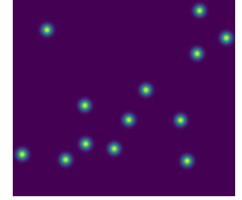 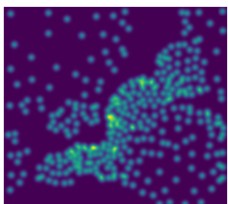

(a) Original image     (b) GT density (from 1st set)     (c) GT density (from 2nd set)

Figure 1: The figure shows one of the cases where the two provided ground truth density map sets do not agree. Image (a) shows the color image, (b) shows the ground truth density map from the first provided set, and (c) shows the ground truth density map from the second set. On (b) the object of interest are balloons, while on (c) the objects of interest are dots on the balloons.

In the original paper, the authors compare FamNet to some common object detectors - Faster R-CNN [15], RetinaNet [10], and Mask R-CNN [7], which were pretrained on COCO data set [9]. Thus they select a subset of FSC-147, which contains categories that also appear in COCO. We manually try to find the intersecting categories and find that 17 categories appear under the same (or similar) name in both data sets. The authors include all images from FSC-147 from those categories in COCO-Val and COCO-Test splits that they provide, and do not leave out any categories that might appear in FSC-147 and COCO.

### 3.2.2 Resizing of FSC-147 images before using FamNet

The authors provide a link to FSC-147 data set in their GitHub repository. However, the images there are already resized as a part of preprocessing before using FamNet. The authors decided to resize all images to a fixed height of 384 pixels. They claim that they adjusted the width of the images in the way that the aspect ratio is preserved. As the authors provide the information about the original dimensions of each image, we checked, whether all processed images are correctly resized to have a height of 384 pixel and if their aspect ratio is truly preserved. We found some cases where the aspect ratio was not preserved. We showed such cases to the authors, who replied that they did not preserve the aspect ratio for images with original width of less than 384 pixels. However, the provided example with a corrupted aspect ratio did not have a width smaller than 384.

### 3.2.3 FSC-147 ground truth density map generation

The authors do not provide the code for the generation of ground truth density maps, but rather provide the already pre-computed density maps and only describe the process. While this is beneficial, as it saves computation time, it is somewhat questionable, as we do not get a full insight into how the data set was generated, and cannot verify their claims. An issue has been opened on the authors' GitHub, but they did not provide the code.

We implemented our own code as described in the paper. We used Gaussian smoothing with adaptive window size and estimated the size of the objects from distances between dot annotations and their nearest neighbor. We averaged those distances to obtain the size of the Gaussian window $s_G$. The authors claim that they use the $\frac{s_G}{4}$ as the standard deviation, however, we could not reproduce the results using this value. We obtained the closest results with $\frac{s_G}{8}$ (see Figure 2 for illustrative example). When we asked the authors about the issue, they suggested that large discrepancies might be due to them computing ground truth deviations on larger images, and then downscaling them to the sizes in the data set. However, we still could not reproduce the same results with the suggested approach. This question still remains open and the issue has not been resolved. Our code produces results most similar to the ground truth density maps, though displacement for some points is visible.

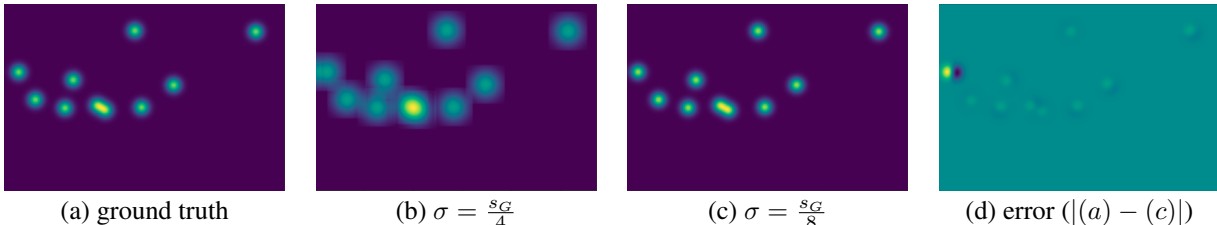

| (a) ground truth | (b) $\sigma = \frac{s_G}{4}$ | (c) $\sigma = \frac{s_G}{8}$ | (d) error ($|(a) - (c)|$) |

Figure 2: The leftmost image (a) represents the ground truth density map, image (b) represents the ground truth density map that we generated by following the authors' description using $\frac{s_G}{4}$ as standard deviation, image (c) represents the ground truth density map obtained with the same process as in (b), but using $\frac{s_G}{8}$ as standard deviation, and image (d) represents absolute point-wise differences between (a) and (c). $\frac{s_G}{8}$ denotes the window size of the Gaussian filter. We can see that (a) and (c) are more similar, while some positional displacements are still noticeable.

#### 3.2.4 CARPK

The authors want to check the performance of FamNet on category specific counting task. They use CARPK data set [8], which contains around 90,000 cars recorded in various parking lots, taken with drones. The data set is already split into train and test set, and for each image ground truths in a form of bounding boxes are provided.

In order to convert the data set into a form suitable for the evaluation of FamNet, we had to create the density maps that represent the ground truths, select the bounding boxes that represent the exemplars, and resize the images to have a height of 384 pixels. The authors do not provide any information about the preprocessing of that data set. We first obtained the distribution for a number of exemplars in FSC-147 data set. We then sampled a number of exemplars $n$ from this distribution for each image and randomly chose $n$ bounding boxes that represent exemplars. To obtain the density maps, we represented each car by a Gaussian filter of the size of the provided bounding box. We set the $\sigma$ of the filter to $\frac{h+w}{16}$ in order to follow the setting of $\sigma$ for FSC-147 data set. However, we did not set the $\sigma$ based on the authors' description in the paper but based on our findings, described in Section 3.2.3.

#### 3.2.5 JHU-CROWD++

As we want to check how the FamNet performs on a typical crowd counting data set, we extend the authors' research and test the pretrained model on JHU-CROWD++ data set [16, 17]. This includes 4327 images collected under a diverse set of conditions (adverse weather, various illumination, varying densities, etc.) and 1.51 million annotations (dots, approximate bounding boxes, etc.).

We again had to do some preprocessing in order to get the data set in the format for evaluation of FamNet. We used the same preprocessing as we did for CARPK data set (see Section 3.2.4).

### 3.3 Hyperparameters

There are several FamNet hyperparameters that have to be set. The authors set $\sigma_G = 8$ used in perturbation loss (see Section 3.1.2) without any explanation why. We therefore decided to check how different values of $\sigma_G$ impact the error of the model. To select the best value of $\sigma_G$ we used grid search. We tested every even integer value between 2 and 20 and report our findings in Section 4.2.1.

We did not test $\lambda_1$ and $\lambda_2$ from adaptation loss (see Section 3.1.3). The authors set their values to $10^{-9}$ and $10^{-4}$, respectively. They say that setting them to such small values is necessary, so the adaptation loss has a similar magnitude to the training loss. We also did not test the number of gradient descent steps and the learning rate during the test time adaptation. The authors said that these two values were tuned along with $\lambda_1$ and $\lambda_2$ on the validation set.

### 3.4 Experimental setup and code

To perform our experiments, we used the authors' and our code. Authors' code is available on their GitHub repository[1] and our anonymized code is available on: `https://anonymous.4open.science/r/`

---

[1]`https://github.com/cvlab-stonybrook/LearningToCountEverything`

re-LearningToCountEverything-51A3. We performed our experiments by running file `test_extended.py`, which is an extended version of the authors' file `test.py`, with some flags for manipulation of different options. To evaluate different values of $\sigma_G$, we used script `choose_sigma.py`, which is run in a similar way as the scripts mentioned before. We also did some preprocessing and testing in our Jupyter notebooks that are self-explanatory to run.

Model is evaluated with absolute error (MAE) and root mean squared error (RMSE), defined as:

$$MAE = \frac{1}{n} \sum_{i=1}^{n} |c_i - \hat{c}_i|, \quad RMSE = \sqrt{\frac{1}{n} \sum_{i=1}^{n} (c_i - \hat{c}_i)^2}, \tag{5}$$

where $n$ denotes the number of instances in test/val set, $c_i$ denotes the number of selected objects on $i$-th image from that set, and $\hat{c}_i$ denotes the predicted count for that image.

### 3.5 Computational requirements

All experiments were ran on GPU only (hence we do not report used CPU and RAM). We ran our experiments on a server with Nvidia Quadro RTX 5000 GPU. Each evaluation of FamNet on test or validation set (FSC-147) takes around 2 minutes without the test time adaptation and around 1 hour and 40 minutes with it. The ablation study with number of exemplars takes around 3 hours (the execution times are shorter when the number of exemplars is decreased). Evaluation of FamNet on subset of categories from COCO data set takes around 40 minutes with adaptation. Evaluation of FamNet on CARPK data set (with adaptation) takes around 1 hour and 20 minutes. We spent around 50 GPU hours to run all of our experiments.

## 4 Results

Our results support the claims of the authors about the quality of their proposed FamNet structure. We managed to reproduce their results exactly (where the code is provided) or up to the point that we can confirm that their model performs as they claim in comparison with the other methods

### 4.1 Results reproducing original paper

We reproduced most of the results obtained with FamNet on FSC-147 and CARPK datasets. The only exception is the ablation study with number of exemplars, where our results do not entirely support the authors' claim.

#### 4.1.1 Evaluation of FamNet on FSC-147 dataset

We managed to get the same results as the authors when testing FamNet on FSC-147 validation and test set, which supports the claim that FamNet outperforms other tested few-shot approaches. The results are given in Table 1 of the original paper. However, we did not test the few-shot methods that FamNet is compared to.

#### 4.1.2 Comparison with object detectors

We tried to reproduce the comparison of FamNet with object detectors, trained on COCO data set. The authors compare FamNet with the detectors on images from FSC-147 data set from categories that overlap in FSC-147 and COCO. We did not manage to reproduce the exact results obtained by the authors (Table 2 in their paper), but we get the results that are within the standard error of theirs or, in case of RetinaNet worse than theirs, and still support the claim that FamNet beats listed object detectors. Our results are shown in Table 1. Additionally to the authors, we report the standard error of MAE estimate, which was calculated from standard deviation. We obtained those results using TorchVision models. The authors use Detectron2 models instead, which perform worse in our experiments.

#### 4.1.3 Number of exemplars ablation study

We reproduced the experiment, that tested the impact of the number of exemplars on the performance of FamNet. However, our results (see Table 2) do not entirely support the claim of the authors that increasing the number of exemplars improves the performance of the FamNet. We can see that by increasing the number of exemplars from 2 to 3, RMSE increased on both test and val set, while the MAE increased on val set and decreased on train set.

Table 1: The results of different object detectors on FSC-147 categories intersecting with COCO categories. Columns SE show the standard error of MAE estimates. Suffixes -Val and -Test to the name of the data set represent the different split of FSC-147 data set that was used.

| | COCO-Val | | | COCO-Test | | |
|---|---|---|---|---|---|---|
| | MAE | SE | RMSE | MAE | SE | RMSE |
| Mask R-CNN (Resnet-50) | 52.04 | 9.61 | 168.23 | 36.66 | 3.31 | 66.58 |
| Faster R-CNN (Resnet-50) | 53.57 | 9.64 | 169.12 | 38.88 | 3.57 | 71.46 |
| RetinaNet (Resnet-50) | 91.17 | 8.75 | 171.85 | 70.2 | 3.34 | 89.85 |
| SSD (VGG-16) | 94.96 | 8.50 | 170.45 | 61.57 | 4.0 | 91.11 |
| FamNet (no adaptation) | 41.13 | 6.32 | 112.92 | 23.23 | 2.42 | 46.79 |
| **FamNet (adaptation)** | **39.82** | **6.04** | **108.15** | **22.76** | **2.37** | **45.92** |

Table 2: The performance of FamNet on FSC-147 data set with respect to the number of exemplars. Columns SE show standard errors of MAE estimates.

| | Val set | | | Test set | | |
|---|---|---|---|---|---|---|
| Number of exemplars | MAE | SE | RMSE | MAE | SE | RMSE |
| 1 | 26.8 | 2.0 | 78.1 | 26.2 | 3.3 | 116.0 |
| 2 | **23.1** | 1.7 | 65.2 | 22.4 | 2.7 | 97.2 |
| 3 | 23.7 | 1.8 | 69.3 | **22.0** | 0.8 | 99.3 |

#### 4.1.4 Evaluation on a category-specific data set

The authors evaluate FamNet on a CARPK data set. We reproduced their results of FamNet trained on FSC-147, but we did not try to reproduce the results for FamNet trained on CARPK. As the authors do not describe the used preprocessing, we did not get exactly the same results as they did. However, our results are close to theirs and still support their claim that FamNet performs well on this category-specific data set. We got MAE **27.9** (with standard error **1.1**) and RMSE **36.4**, while the authors got MAE **28.8** and RMSE **44.4**.

### 4.2 Results beyond original paper

Additionally, we tested how $\sigma_G$ (Section 3.1.2) and correction of the Min-Count Loss affect the model's performance, evaluated the model on another category-specific data set, visually inspected the errors of the model and effects of test time adaptation.

#### 4.2.1 Impact of $\sigma_G$ on the error of the model

Since the authors do not provide any justification for setting $\sigma_G = 8$, we test how the MAE of FamNet changes with different $\sigma_G$ (see Figure 3). We can see that $\sigma_G$ has practically no impact on MAE of FamNet.

#### 4.2.2 Min-Count Loss correction

Since we have noticed that the authors' definition of Min-Count Loss differs from their implementation (see Section 3.1.1), we tested how it affects the error of the model. Our results did not show any significant difference in MAE and RMSE.

#### 4.2.3 Evaluation on JHU-CROWD++

To test the performance of FamNet on category-specific data set even further, we evaluated it on the JHU-CROWD++ data set (see Section 3.2.5). We use a model trained on FSC-147. The results are shown in Table 3. We can see that FamNet performs worse than baselines. However, this data set is challenging (large number of objects, small bounding boxes) and training the model on that data set with a higher number of exemplars would likely boost the performance.

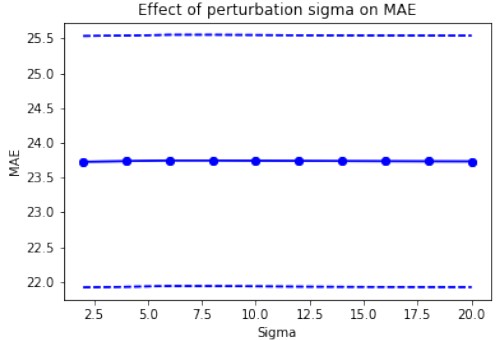

Figure 3: MAE (full blue line) of FamNet depending on $\sigma_G$ and the standard error of the estimate (dotted line). The tested values were even integers between 2 and 20.

Table 3: Evaluation of FamNet on JHU-Crowd++, trained on FSC-147, compared to two other models evaluated on the same data set. Column SE shows the standard error of MAE estimate.

|  | MAE | SE | RMSE |
|---|---|---|---|
| MCNN | 188.9 | / | 483.4 |
| CSR-Net | **85.9** | / | **309.2** |
| FamNet (our results) | 256.9 | 15.0 | 652.5 |

#### 4.2.4   Performance of the model without adaptation

Additionally, we visually inspect the images, where absolute error, normalized by the ground truth count, is the highest or the lowest. Our observations and visualisations are described in Appendix B.

#### 4.2.5   Effect of adaptation on model's predictions

To inspect the effect of adaptation, we analysed the most positive and negative effects of adaptation on model's performance. We describe the results in Appendix C.

### 5   Discussion

We tried to reproduce the results from the paper *Learning to count everything*. We obtained the same results as in the paper for some experiments. For others, our results are still close enough to the papers'. We confirmed that FamNet outperforms other few-shot approaches when it comes to object counting and that FamNet performs well even on a category-specific data set. Our experiments disprove the authors' claim that increasing the number of exemplars decreases FamNet's error. We assume that this is due to the fact that we discarded different exemplars than the authors. This might suggest that choosing correct exemplars is more important than choosing more of them.

#### 5.1   What was easy

A demo app with clear instructions helped with the understanding of the model. The model's architecture was understandable from the code and the paper. It was easy to reproduce the results on FSC-147 with a pretrained model.

#### 5.2   What was difficult

Reproducing the ground truth density maps was difficult, as the process in the paper did not lead to the paper's results, and the code for it was not provided. We did our best to mimic the ground truth density maps. Evaluating other models for which the code was not provided was challenging, as no parameters were given (e.g., confidence or intersect over union thresholds for object detectors). We struggled obtaining a good enough GPU. Due to lack of time, we were unable to train the model ourselves, and we delegate this to future work.

#### 5.3   Communication with original authors

We contacted the authors three times through issues on GitHub. They were helpful and responsive, but we have not resolved all of the issues.

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

## A    Input features to density prediction module

The authors use a density prediction module that is agnostic to the visual categories. Instead of feeding the features obtained from the feature extraction module directly, they rather use the correlation map between the exemplar features and image features as the input to the density prediction module. In Figure 4 we show an example of an input (correlation features) to the density prediction module.

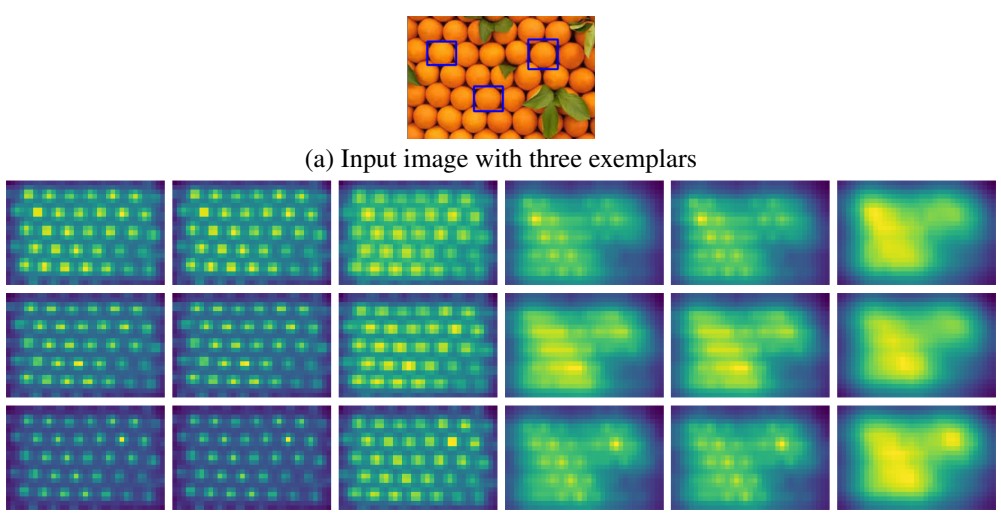

(a) Input image with three exemplars

(b) Correlation maps

Figure 4: Figure shows an example of an input image (a) from FSC-147 with three exemplars (blue rectangles), and correlation maps between exemplar and image features (b), that are fed to the density prediction module. Each row corresponds to one exemplar, while each column corresponds to one combination of scale (3 scales) and feature output from third or fourth block of ResNet-18.

## B    Images with the best and the worst relative MAE on test set without adaptation

We visually inspect the images where absolute error normalized by the ground truth count is the highest or the lowest, and show some of the images in Figure 5. We can see that the algorithm predicts density maps with highest relative count error in cases where he predicts counts for wrong objects. In all three cases defined shapes which confuse the algorithm are present. Algorithm works the best on images, where the shape of the object it counts is well-defined and differs from the background, or there is no background at all. Adaptation in some cases improves the prediction, while in some cases it makes it worse. Thus, we investigate the affect of adaptation in the next section of appendix.

## C    Effect of adaptation on predictions

We inspect on which images the absolute error normalized by the ground truth count is improved or worsened the most. We show examples of those images in Figure 6. We do not observe any special pattern in the shown images. The main reason for a bigger impact of adaptation on those images is that their relative errors were quite high/low and consequently absolute changes in the prediction had a bigger impact on relative error.

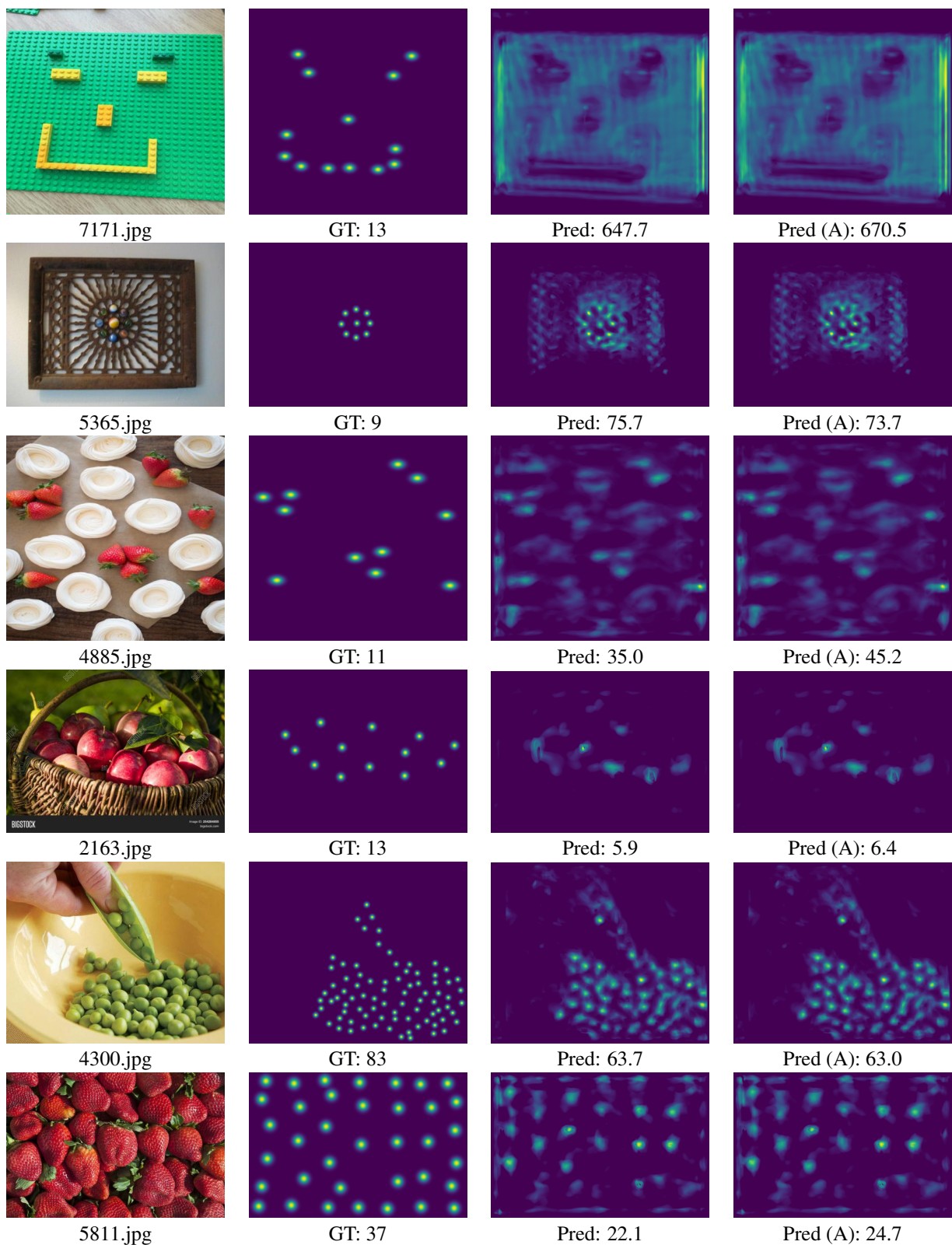

Figure 5: The first column represents input images, the second represents ground truth density maps, while the third and the fourth represent predicted density maps without and with test-time adaptation, respectively. The first three rows include cases where absolute error normalized by ground truth count is among the highest in the test set, while the last three rows where it is among the lowest.

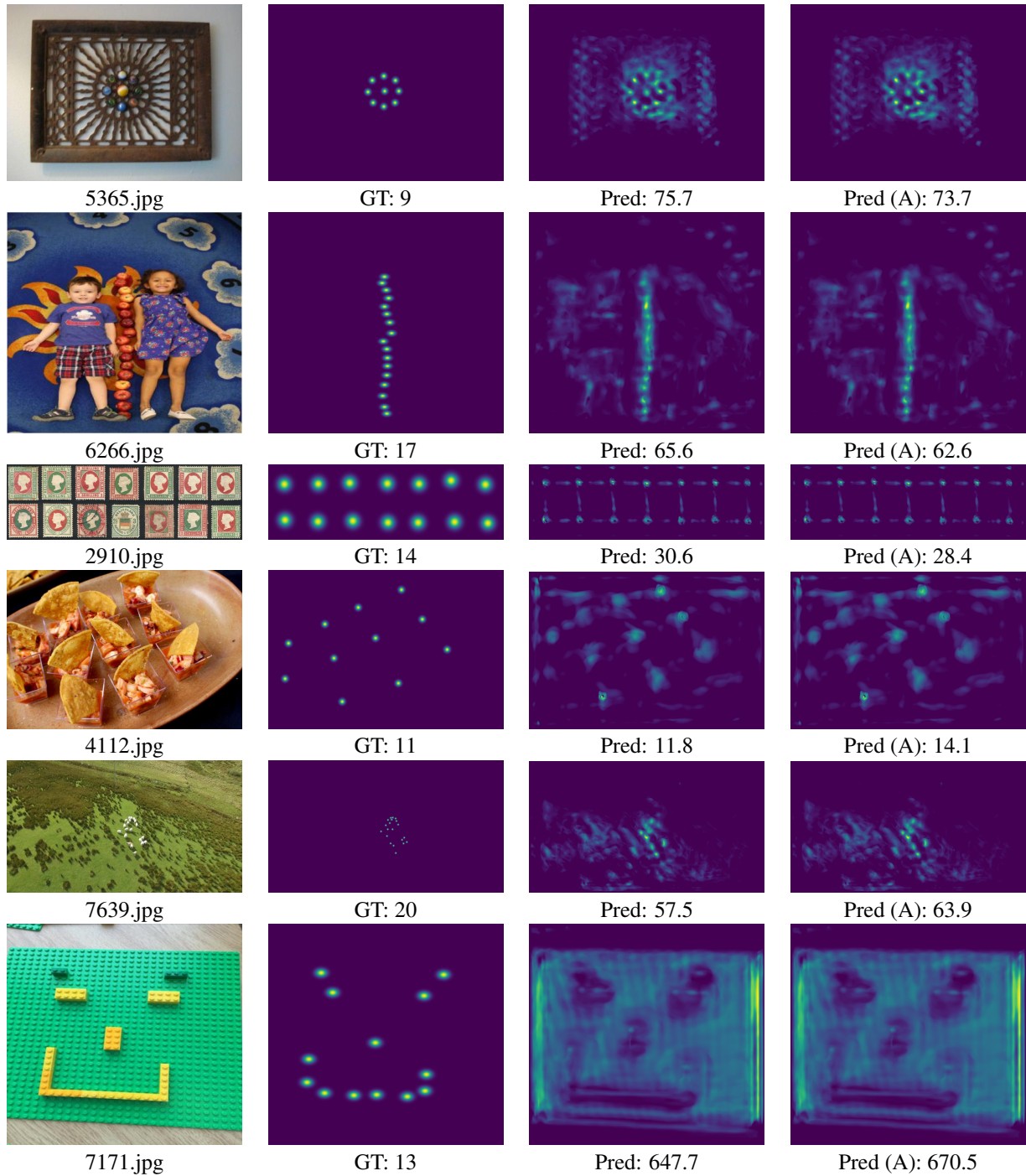

Figure 6: The first column represents input images, the second represents ground truth density maps, while the third and the fourth represent predicted density maps without and with test-time adaptation, respectively. The first three rows include cases where the test time adaptation decreased relative error, while the last three rows include cases where the test time adaptation had a negative impact.

