# OpenReview forum: "[Re] Learning to count everything"
_ML_Reproducibility_Challenge/2021/Fall — RC2021 OutstandingPaper_

### Official Review · Reviewer_T3ho · 2022-03-05
**Confirms Limited Reproducibility**

**Rating:** 8
**Confidence:** 5

**Review:**

The review claims that the training could not be reproduced because of limited information in the paper, and supplementary information provided by the authors. This is valid.

Otherwise, the results of the paper seem to have been reproduced approximately, because the proposed model analogously outperforms the baseline.

The reproducibility report is OK.

---

### Official Review · Reviewer_217e · 2022-03-16
**Great analysis, would benefit from retraining models on mentioned datasets**

**Rating:** 7
**Confidence:** 4

**Review:**

1. The authors do a thorough analysis with the pre-trained network on additional datasets such as CARPK, JHU-CROWD++. They discuss the impact of different hyper-parameters, especially for constructing the ground truth density maps.
2. The authors investigate the release code to evaluate if it matches the paper description. Additionally, they evaluate the effect of MAE vs RMSE loss on min-count on the final performance.
3. The only weakness of the report is utilizing the pre-trained model. In that respect, the authors haven't really evaluated if the provided training code works. This could be a reason why the obtained results on JHU-CROWD++ are worse than baselines and strengthen the impact of this reproducibility report.

Minor:
1. It would be great to make the report more self-contained by concretely stating how MSE loss is computed (in line 98). For example, writing the loss as an equation and describing density maps would be really helpful here. Presently, the report just assumes that the reader knows about density maps which may not be the case.

---

### Meta-Review · Area_Chair_ESXX · 2022-04-09

**Recommendation:** Accept (Outstanding Paper)
**Confidence:** 4

**Metareview:**

Reviewers uniformly praised the work, highlighting that authors showed that the original paper didn't have enough information for a reproduction (which as in interesting and useful result). They also praised that the hyperparameter search.

---

### Decision · Program_Chairs · 2022-04-09

**Decision:**

Accept (Outstanding Paper)

**Comment:**

Following the recommendation of reviewers and meta-reviewer, the paper is accepted for ML Reproducibility Challenge 2021, and will be published in the upcoming special edition of ReScience Journal.

Additionally, after several rounds of discussion and incorporating recommendations from the Area Chairs and Program Chairs, the report has been granted an **Outstanding Paper Award** due to its exceptional quality of all-round reproducibility effort. Congratulations!